# Problems Associated with Deprescribing of Proton Pump Inhibitors

**DOI:** 10.3390/ijms20215469

**Published:** 2019-11-02

**Authors:** Holmfridur Helgadottir, Einar S. Bjornsson

**Affiliations:** 1Faculty of Medicine, University of Iceland, 101 Reykjavik, Iceland; hofihelgad@gmail.com; 2Division of Gastroenterology and Hepatology, The National University Hospital of Iceland, 101 Reykjavik, Iceland

**Keywords:** proton pump inhibitors, gastrin, rebound acid hypersecretion, deprescribing, step-down, tapering, on-demand, discontinuation

## Abstract

Proton pump inhibitors (PPIs) are recommended as a first-line treatment for gastroesophageal reflux disease (GERD) and other acid related disorders. In recent years, concerns have been raised about the increasing prevalence of patients on long-term PPI therapy and inappropriate PPI use. It is well known that short-term PPI therapy is generally well tolerated and safe; however, their extensive long-term use is a major global issue. One of these long-standing concerns is PPI-induced gastrin elevation secondary to hypoacidity. Hypergastrinemia is believed to play a role in rebound hyperacidity when PPIs are discontinued resulting in induced dyspeptic symptoms that might result in the reinstitution of therapy. Gastrin exerts tropic effects in the stomach, especially on enterochromaffin-like (ECL) cells, and concerns have also been raised regarding the potential progression to dysplasia or tumor formation following long-term therapy. It is well known that a substantial number of patients on long-term PPI therapy can discontinue PPIs without recurrence of symptoms in deprescribing trials. What is unknown is how sustainable deprescribing should be undertaken in practice and how effective it is in terms of reducing long-term outcomes like adverse drug events, morbidity and mortality. Moreover, there is no clear consensus on when and how deprescribing strategies should be attempted in practice. This review sought to summarize the harms and benefits of long-term PPI therapy with special focus on gastrin elevation and its relation to deprescribing studies and future interventions that may improve PPI use.

## 1. Introduction and Gastric Acid Regulation

The key player in acid secretion is a H^+^/K^+^ ATPase (the proton pump) located in the parietal cell which is responsible for the transport of hydrogen ions into the gastric lumen [1]. The H^+^/K^+^ ATPase has been shown to be an α, β-heterodimeric enzyme which catalyses a one-to-one exchange of hydrogen (H^+^) and potassium (K^+^) ions [2,3]. The parietal cells bear receptors for three stimulators of acid secretion, acetylcholine, histamine and gastrin reflecting neural, paracrine and endocrine control, respectively (Figure 1). Gastrin is the main mediator released from antral G-cells into peripheral blood (i.e., hormonal pathways), stimulating enterochromaffin-like (ECL) cells via gastrin/cholecystokinin_2_ (CCK_2_) receptors accelerating histamine release [1,4]. Histamine diffuses (i.e., through a paracrine pathway) to interact with H_2_-receptors on parietal cells. Vagal efferent releases acetylcholine that also stimulates the parietal cells directly by binding to M_3_ receptors. It is now recognised that the stimulation of acid secretion by gastrin mainly occurs via histamine release from the ECL cells, the gastrin–ECL axis. Histamine from the ECL cells is considered the limiting step in stimulating maximal gastric acid secretion [5]. Gastrin/CCK_2_ receptors have been well documented on ECL cells but not on parietal cells [6]. However, gastrin is believed to stimulate parietal cells directly, to some extent, but this is considered to be a less extensive mechanism [1]. The stimulatory effect of acetylcholine and gastrin is mediated by an increase in cytosolic calcium (Ca^2+^), whereas that of histamine is mediated by the activation of adenylate cyclase and generation of cyclic adenosine monophosphate (cAMP) [2]. The second messengers (cAMP and Ca^2+^) activate protein kinases and phosphorylation cascades that ultimately activate transport of the proton pump (H^+^/K^+^ ATPase) from cytoplasm to the canaliculus where the pump can exchange intracellular H^+^ with the extracellular K^+^, which represents the gastric acid secretion [2] (Figure 1). Therefore, a combination of the effects of these stimulants can augment gastric acid secretion. The main inhibitory factor is somatostatin released from oxyntic and antral D-cells. Intragastric acidity regulates gastrin release through a negative feedback inhibition, whereby acidity stimulates antral D-cells to release somatostatin but food or neutral stomach content inhibits their secretion of somatostatin [7]. It is now recognized that the stimulation of acid secretion by gastrin mainly occurs via histamine release from the ECL cells, the gastrin–ECL axis (Figure 2).

PPIs are a class of medications that selectively and irreversibly inhibit the proton pump that accomplishes the final step in acid secretion. PPIs are weak bases and therefore accumulate in the acidic space, the secretory canaliculus, of parietal cells [3]. After acid-induced activation PPIs covalently bind to the active proton pump (H^+^/K^+^-ATPase); the binding is achieved through the disulphide bond between the activated PPI and cysteines of the pump enzyme [3]. This covalent binding enables the prolonged inhibition of acid secretion, even after the drug concentration in blood has waned. The duration of the inhibitory activity is variable and affected by pump turnover and the loss of covalently bound PPIs [8]. PPIs are most effective when the parietal cells are stimulated to secrete acid, as they are after a meal; as such, PPIs should be administered before a meal [9]. Acid inhibition by PPIs decreases gastric acid levels without the disruption of G-cells. The interruption of this feedback mechanism leads to augmented gastrin release and gastrin elevation in the blood [10]. It is widely accepted that all patients undergoing long-term PPI therapy develop some level of gastrin elevation, to varying degrees, but only a small portion develop hypergastrinemia, defined as gastrin levels being above the upper limit of the reference range for fasting blood gastrin [11,12,13]. Hypergastrinemia has become a topic of research and many studies have raised concerns about the clinical significance of continuous gastrin elevation in patients on continuous therapy.

PPIs are the mainstays in treating many acid-related disorders, such as gastroesophageal reflux disease (GERD), peptic ulcer disease (PUD) and dyspepsia (acid related), as a part of eradication of *Helicobacter pylori* (*H. pylori*) with two antibiotics and hypersecretory conditions (e.g., Zollinger-Ellison syndrome) [14]. They are also used as ulcer prophylaxis in patients with history of PUD, in critically ill patients in the Intensive Care Unit and people who use nonsteroidal anti-inflammatory drug (NSAIDs) [14]. While PPIs are effective and have a relatively desirable safety profile there is still uncertainty in regards to the safety of long-term, and often life-long, PPI therapy. The last decade has seen some major changes in the landscape of PPI use, a growing number of long-term users with and without adequate indication, increasing number of reports about adverse events due to long-term use and the implementation of PPI deprescribing to address these issues. The current review is focused on secondary hypergastrinemia and acid rebound following discontinuation of PPIs and their role in deprescribing studies as well as future directions.

## 2. Safety of Long-Term PPI Use

In recent decades, the prevalence of proton pump inhibitor use has steadily increased [15,16,17]. PPIs are commonly continued long-term although the most common indications such as GERD and mild esophagitis are recommended for short-term use (e.g., eight weeks) to heal inflammation and resolute symptoms in management guidelines [18]. Concerns about the appropriateness of their use and potential adverse effects that might stem from continuous PPI therapy have been growing [19]. There is also inappropriate prescribing of PPIs and in cross-sectional studies only approximately 30% of people were prescribed PPIs with appropriate indication concordant with guideline recommendations [20,21,22]. PPIs’ clinical efficacy and that they are generally well tolerated is likely the reason for their overutilization and inappropriate use but concerns regarding their long-term safety are increasing. PPI overuse (e.g., poor indication, excessive dose, excessive duration) contributes to polypharmacy and potential risk of drug interactions and side effects. Last but not least there are economic implications of PPI overuse.

The putative side effects of PPI therapy can be divided into four categories based on the effects and metabolism of this drug class:

(1) Hypochlorhydria: Side effects related to the direct effect of acid suppression or PPI-induced hypochlorhydria has been associated with an increased risk of infections [23,24,25,26,27,28], malabsorption of nutrients, minerals and vitamins and increased risk of osteoporotic fractures [4,25].

(2) Idiosyncratic: Idiosyncratic side effects are rare and as the term implies are unpredictable among PPI users. These include renal failure (i.e., acute interstitial nephritis) [29] and the association with cardiovascular disease [30] and dementia [31].

(3) Pharmacokinetic (PK) interaction: The PK interaction is related to the effects on the cytochrome P450 (CYP450) cytochromes and possible drug interactions when prescribed with other drugs. There is lack of convincing clinically important drug interactions with PPIs, the most studied combination is the use of clopidogrel with PPIs [32], which has been associated with higher adverse cardiovascular events; however, results have been mixed and it is unclear if there is a true difference in clinical outcomes [33].

(4) Hypergastrinemia: [4,23,24,25,26,27,28] The clinical implications of PPI-induced hypergastrinemia are also of concern. These concerns are largely related to gastrin’s potent trophic effects. The rebound acid hypersecretion (RAHS) phenomenon is believed to result from gastrin’s hypertrophic effects on ECL cells in the stomach (Figure 3), leading to increased acid production even after therapy has been discontinued (Figure 4) and rebound symptoms and possibly driving the cycle of inappropriate prescribing [5,34]. Another concern is that long-term gastrin elevation has a gastric carcinogenic effect [35]. PPI treatment induces ECL-cell hyperplasia and can provoke gastric polyp formation [36]. The subsequent development of ECL carcinoids and carcinomas has been described in numerous case reports [37,38,39]. Furthermore, population-based studies have shown an increased risk of gastric cancer in patients with long-term PPI use [40,41]. It has been disputed whether gastrin alone is sufficient to induce gastric cancer or if it acts as a co-factor once premalignant changes are triggered. While a causal relationship between PPIs and neoplasia has not been established in humans, it remains advisable to avoid high serum gastrin values for prolonged periods.

While these long-term side effects are uncommon and associations are mainly based on observational studies without a clear causality, with increased long-term PPI use more people are at risk. The fact that annual incidence remains stable across time while prevalence increases [15,16] makes problems associated with discontinuation of PPIs more likely than increased morbidity or new indications driving the increase in PPI use. PPIs are prescribed in a wide range of conditions but the increase is most evident among the elderly [16] but they are often prescribed to children and childbearing women [42]. When PPIs are inappropriately prescribed or continued beyond the recommended course of treatment they are very unlikely to provide benefit. A cross-sectional analysis of 901 Danish primary care patients on long-term PPIs demonstrated that 454 (51%) had uninvestigated symptoms and 200 (22%) were treated despite a normal upper GI endoscopy [22]. While treatment with PPIs is initiated mainly in primary care, opportunities to improve PPI use exist in primary-care and hospital patients. Inappropriate PPI initiation rates in hospitals are also high, ranging from 40–81% on general medicine wards [20]. In a German cross-sectional observational study 54% (371/681) of patients had inadequate indication for continuous PPI medication in their discharge letters [20].

## 3. Definition of PPI Deprescribing

There is no uniform definition of deprescribing PPIs; it involves the process of reducing and/or stopping the PPI therapy after consideration of therapeutic indication, benefits and risk. The aim of deprescribing PPIs in most cases is to reduce medication burden and potential adverse effects while maintaining quality of life [43]. PPI deprescribing algorithms have been proposed and published in Canada [43] and Australia [44]. Both recommend deprescribing of PPIs in adults who are symptom free after a minimum of four-week PPI therapy for GERD or upper GI-symptoms. However, deprescribing can be difficult, and there is no evidence-based method of stopping or reducing PPIs. Various different approaches to deprescribing have been outlined in prior trials and guidelines and Figure 5 demonstrates different approaches in a flow chart and Table 1 lists the main steps in deprescribing PPIs. In the current era of PPIs recommendations are coming out to advocate for PPI deprescribing in cases where they might no longer be needed [45], but despite this there seems little change in prescribing practice and sustained behaviour change has not been achieved.

## 4. Rebound Acid Hypersecretion

The highest relapse rates seem to be in the abrupt discontinuation studies [46,47], this might be affected by the expected and described rebound effect. The development of dyspepsia in healthy volunteers after the discontinuation of PPIs, in comparison with controls receiving placebo, has been associated with gastrin elevation induced by the acid-suppressive drugs [48,49]. A previous study did not find significant gastrin elevation after five days of standard-dose PPI therapy [50] but a recent study showed a significant increase in gastrin levels after only four days of PPI intake in healthy volunteers [51]. The increase in gastric acid secretion to above pre-treatment or baseline levels after withdrawal from PPIs has been well documented in several physiological studies [52,53]. However, the significance of RAHS in patients with gastroesophageal reflux disease has not been as well investigated and is therefore less clear. A study in GERD patients after withdrawal from five-day PPI therapy did not show evidence of RAHS [54], nor did on-demand therapy with PPIs in endoscopy-negative GERD patients [55]. A recent study on *H. pylori*-negative patients with erosive esophagitis (EE) randomised to a placebo after experiencing the healing of EE after four or eight weeks of PPI therapy did not demonstrate any indication of recurring heartburn symptoms worsening beyond baseline levels within two months of discontinuing PPIs [56].

Serum gastrin was found to be an independent predictor of PPI requirement in the Swedish study on the discontinuation of PPIs after long-term treatment [11]. Longer duration of PPI therapy has also been shown to be associated with a lower likelihood of being able to step down from twice-daily to once-daily PPI therapy in GERD patients, supporting the contention that longer-term therapy may be associated with hypergastrinemia-induced hypersecretory capabilities [57]. High quality studies on RAHS in patients after long-term PPI therapy are largely lacking. RAHS is believed to contribute to difficulties in the discontinuation of treatment and acid rebound, which might explain, at least in part, the increase in long-term users without an adequate indication of needing PPIs [58]. Concerns have been raised about this physical dependence upon PPIs and it has been hypothesised that the increase in the incidence of GERD over the recent decades might be due to the worsening of reflux symptoms caused by RAHS. In other words, PPI therapy for GERD might be worsening the disease itself [34]. Gastrin elevation is most prominent in the first few months and up to 1–2 years of long-term treatment [59,60]. Thus, it seems imperative to minimize the duration of treatment before deprescribing, as that could substantially reduce the risk of acid-rebound. Tapering is also believed by some researchers to be a more successful approach to decrease the gastrin elevation before discontinuation [61]. Studies have reported a dose-dependent gastrin response induced by PPI therapy with higher levels in patients on higher PPI doses in mg [62] and with higher frequency of PPI intake (daily vs. every other day) [50]. Furthermore, a positive association has been observed between PPI dose expressed in dose per weight (mg/kg) and gastrin levels [13].

While symptoms can, in many patients, rapidly reoccur after treatment discontinuation, only 10% (11/113) of patients on long-term PPIs for dyspepsia in general practice in the Netherlands stated that their physician had discussed when to discontinue PPIs [63]. The same report revealed that a simple patient education intervention could reduce their use of PPIs where 24% of patients stopped or reduced their dose at three months compared with 7% in the control group [63]. A cross-sectional analysis among Danish primary care patients on long-term PPIs revealed that 61% (119/194) of patients had previously attempted to withdraw therapy but 39% (75/194) had never attempted discontinuation [22]. Based on experience from studies on healthy volunteers [48,49] patients should also be informed that possible breakthrough symptoms during the first weeks after PPI discontinuation can be rebound symptoms that will cease and can be treated with alternatives like antacids or H_2_RAs instead of leading to resumption of PPI therapy.

## 5. PPI Desprescriping Trials

The goal of deprescriding or discontinuing PPIs is to reduce medication burden and possible harm of continued therapy. The most common approch by patients is on-demand therapy as many patients do not take them daily as prescribed [64]. A systematic review for the Cochrane Collaboration of randomized controlled trials published between 2003 and 2016 included six studies; five of them assessed deprescribing via an on-demand approach among patients with non-erosive reflux disease or milder form of EE (LA grade A to B) [65]. Pooled data from these studies showed that approximately 84% tolerated the intervention, although there was a significant difference in relapse rate compared with the maintenance group (16% vs. 9%, *p* < 0.0001) [65]. The only study that included an endoscopy after follow-up found that 5% of NERD patients in the on-demand group developed mild esophagitis compared with none in the continuous group (*p* < 0.0001) [66]. Despite their higher symptom load, on-demand therapy was well tolerated and around 80% were satisfied with the on-demand treatment and there was a significant reduction in medication burden measured in pill use [66,67,68]. Two of the six studies also assessed abrupt discontinuation. One with a three-month placebo arm with rescue medication where 19% of GERD patients on long-term PPI therapy terminated treatment [47] and the other one evaluated abrupt discontinuation in the elderly (>65 years) after six months PPI therapy for acute esophagitis [46]. Pilotto et al. demonstrated a high relapse rate in the elderly in the following six months with 80% relapse in the PPI withdrawal group compared with 30% in the maintenance group [46]. While this study did not support abrupt discontinuation in the elderly it did include a step-down after the first eight weeks of treatment from standard-dose to low-dose pantoprazole in those with healed esophagitis and continued to observe high healing rates, 80% after 12 months of low-dose maintenance therapy. These data support that low-dose PPI therapy might be sufficient to reduce relapse of esophagitis.

Inadomi et al. analyzed the success of a step-down approach among patients with symptoms of heartburn or acid regurgitation and found that nearly 80% of participants could be successfully managed with lower doses of PPIs [57]. An RCT conducted in Iceland among GERD patients with endoscopically proved EE also found that step-down management was possible in 76% of patients on long-term PPIs [60]. The reported results of successful step-down studies vary from 50% to 88% in unselected GERD patients [57,69,70,71]. Lowering the daily dose is an important maintenance therapy strategy to reduce the rate of unnecessary high-dose PPI therapy and can be the first step in stopping PPI therapy via tapering. The deprescribing and tapering of PPI treatment, rather than abruptly stopping their administration, has been suggested to minimize withdrawal symptoms, particularly in patients who have been treated for longer duration and those who have experienced symptom recurrence after PPI withdrawal. However, data to support the hypothesis that tapering reduces withdrawal symptoms are lacking. In one Swedish study, tapering was conducted over a period of three weeks before discontinuation and compared with abrupt discontinuation [11]. Tapering was not shown to be superior for the successful discontinuation of PPI therapy, 31% in the tapering group vs. 22% in the non-tapering group were off PPIs at one-year follow-up [11].

Table 2 summarises characteristics of nine deprescribing trials published between 2003 and 2017. Despite heterogeneity in study design the success rate of deprescribing among GERD patients increased from abrupt discontinuation to step-down to on-demand intervention, from 25–62% [11,46,47,72] to 76–80% [57,60] to 69–92% [66,67,68]. Factors that where associated with or predicted PPI requirement were longer PPI duration [57], male gender [60,72], GERD symptom severity [11] and gastrin elevation [11]. It is important to measure the quality of life in patients when monitoring PPI therapy. The aim of deprescribing is not complete resolution of symptoms. Occasional episodes of heartburn and acid regurgitation are normally found in a healthy population and can be associated with certain dietary patterns. Studies and surveys on the prevalence of reflux symptoms have demonstrated that approximately 40% of adults from the general population suffer from reflux symptoms in Western countries [73,74,75]. Thus, complete symptom relief as an endpoint in deprescribing trials may not be what the patients aim for long-term when they are informed and might also fear “addiction” and long-term side effects of PPIs. On-demand PPI treatment satisfies the majority of GERD patients [66,68]. In practice many patients prescribed daily therapy take their PPIs on-demand and adopt some kind of lifestyle measures (e.g., avoidance of food and beverages that produce symptoms) [64].

## 6. Deprescribing Guidelines

Recommendations for deprescribing PPI therapy are developed to provide useful guidance for both patients and clinicians to make decisions about when and how to deprescribe PPIs [43]. The Australian algorithm produced by the National Prescribing Service involves a stepwise approach for patients with GERD where symptom control guides the step-down or step-up necessary for reflux symptom control [44]. The Australian algorithm recommends gradually reducing the dose before stopping to manage RAHS while the Canadian algorithm recommends different deprescribing approaches (e.g., decrease to lower dose, stop and use on-demand or stop abruptly) without favoring an optimal approach [43]. The Canadian algorithm also recommends a follow-up at four and 12 weeks and both algorithms include lifestyle changes and alternative medications (e.g., antacids or H_2_RAs) to manage occasional mild symptoms [43,44]. Figure 5 demonstrates the most common approaches of PPI deprescribing but there is no international consensus on the best approach to discontinue PPIs. Nevertheless, these guidelines might be a useful reminder for clinicians to aim for discontinuation of therapy if the indication for PPIs is not strong since long-term PPI therapy without appropriate monitoring and reassessment seems to be commonplace.

PPIs are widely prescribed today, not only for acid-related disorders but also, frequently, for a variety of upper GI conditions not necessarily related to acid, partly due to a lack of other therapeutic modalities for upper GI symptoms. Thus, the focus on deprescribing should primarily be on avoiding unnecessary long-term use and using of the lowest effective dose of PPIs when indicated. The authors would like to suggest two important practice points when prescribing PPIs, the preventive approach and the detective approach. The preventive approach is when empiric PPI trial is used as a diagnostic indicator for GERD in patients with typical reflux symptoms in the absence of alarm features. The trial should be a short PPI course, we believe the 4–8 weeks treatment approach that is often prescribed is more than what is needed to assess treatment response. It is estimated that the daily dosing of PPIs reaches a steady state of inhibition after five days, and that the state is the inhibition of about 66% of the maximal acid output [3]. Therefore, a short one-week PPI course should be enough to assess response. There is need for randomized trials to support this but this approach is likely to avert overutilization and minimize possible withdrawal symptoms in patients who do not benefit from PPI therapy. The detective approach is when renewing PPI prescriptions for patients already on PPI therapy. Then the steps listed in Table 1 should be used to assess the patient’s need for continued PPI therapy and candidates for deprescribing should subsequently attempt to reduce dose and/or frequency of PPIs or stop them. The success and relapse of the deprescribing approaches listed in Figure 5 should be periodically reevaluated so that the lowest effective PPI dose is used. This approach is likely to lower medication costs and lower the risks of long-term side effects among GERD patients prescribed maintenance PPI therapy for symptom control.

With the emerging interventions to minimize PPI overuse it is important to mention that there can also be PPI underuse. Long-term PPI therapy can be appropriate and where there is a clear indication the benefit outweighs potential risk. Patients with Barretts esophagus (BE), severe esophagitis grade C or D, history of bleeding GI ulcer or bleeding risk with chronic NSAID use are recommended to continue PPIs or consult a gastroenterologist before discontinuation [18]. The ACG guidelines recommend long-term PPI therapy for patients with severe grade EE since they have a high rate of relapse after PPI therapy is discontinued [76] and BE due to increased risk of dyplasia and/or adenocarcinoma [77]. There are more indications for continuous PPI therapy, which can be found in Table 3 [78].

## 7. Large Gap in the Knowledge of What Should Be Recommended

There is lack of evidence to determine optimal deprescrbing approaches (e.g., abrupt discontinuation, tapering to the lowest effective dose or use of alternative therapy to overcome potential rebound symptoms). Due to the paucity of published deprescribing trials, the current evidence is generally of low methodological quality and provides relatively low-certainty evidence. Randomized controlled trials (RCT) are considered to be the gold standard when it comes to high levels of evidence. However, RCTs are not always applicable when estimating the safety and effectiveness of drug deprescribing, as in the case of PPIs, when the adverse effects are rare or take a long time to develop. Furthermore, the small sample size often seen in the RCTs tends to be relatively homogeneous. The guidelines published are mainly based on studies on patients with moderate GERD or mild esophagitis. Thus, they are not always adequately representative of the real-world population of PPI users. Furthermore, whether deprescribing approaches success might differ between different patient groups being targeted. There is also a need for assessment of the cost benefit and additional physician visits following deprescribing.

In most deprescribing studies the majority of patients have tolerated the intervention but the duration of follow-up has been limited. Most trials mentioned above were of short duration (up to 12 months) and very limited data are available on long-term benefits and side effects experienced following deprescribing (i.e., rates of recurrence of esophagitis, GI-bleeding, strictures, GI-cancer, hospitalization, death, fractures, *Clostridium difficile* infections, hypomagnesemia, etc.). It is, for example, well known that patients with grade C and D esophagitis have higher relapse rates and have therefore been excluded in deprescribing trials [76]. This remains an important patient group for future research as population-wide educational programs on deprescribing will also reach these patients and might promote underuse or inappropriate discontinuation of PPIs among patients who need maintenance therapy, putting them at risk for consequences like upper GI-bleeding.

There is insufficient data available on the optimal deprescribing approach, e.g., whether patients taper their dose for some time before stopping. While no evidence that PPI step-down before discontinuation is better than abrupt discontinuation [11], tapering is believed to be more effective [61] and this strategy for discontinuation is recommended in the Australian algorithm [44]. However, this recommendation is mostly based on physiological studies [52,53] since there is limited data on PPI withdrawal in patients and with conflicting results [56,57]. Very few studies have attempted to identify RAHS as a relevant phenomenon in patients, future studies on RAHS among patients should focus on when patients can expect rebound symptoms if relevant after PPI withdrawal. It has been shown that the vast majority of patients in deprescribing trials who fail the intervention do so after 3–4 weeks and very few reuptake their prior PPI therapy thereafter [11,57,60].

This kind of information could give both prescribers and patients a time frame to predict when they might expect this phenomenon after stopping treatment. Those with relapse can then be aware of when it occurs and temporarily control the rebound with antacids or H_2_ antagonist during the one, two or three weeks after PPI withdrawal that are associated with RAHS rather than reinstitute unnecessary PPI-therapy.

The lack of quality evidence of serious harm of deprescribing and the benefit of less medication burden and lowering medication cost has been a strong drive for deprescribing guidelines, especially in primary care and long-term care. The increased usage of PPI is higher among older adults [15], a population particularly susceptible to adverse drug events. If the prevalence of PPI users continues to increase worldwide accompanied by the current growth of the population aged 65 and older, more people will be at risk of long-term side effects of continuous PPI therapy.

## 8. Conclusions

Deprescribing PPIs among long-term users has become a topic of research due to the extensive use of PPIs and associated safety considerations. The fact that approximately 30% of patients on long-term can discontinue long-term PPI therapy [11,72] and up to 80% are able to lower the dose [57,60] in deprescribing studies demonstrates that there are several interventions that can be made to decrease the use of PPIs (Table 2). Differences in deprescribing success can be related to different study design, selection of patients with different symptom severity and whether the indication for PPI therapy and dose was appropriate. There is need for high quality, sustainable interventions in deprescribing PPIs. Despite the fact that deprescribing approaches today are based on low to moderate quality of evidence the need for PPI deprescribing is strong and ongoing efforts to identify and address inappropriate PPI continue to be needed. However, “which came first: the chicken or the egg?” PPI overprescribing and misprescribing in practice continues to add to the PPI overuse that has spread like an epidemic across the world. There is also need for preventive measures in clinicians’ daily practice when PPIs are first prescribed, these include a documented indication, treatment duration plan and a set review date to reassess the need for ongoing treatment.

## Figures and Tables

**Figure 1 ijms-20-05469-f001:**
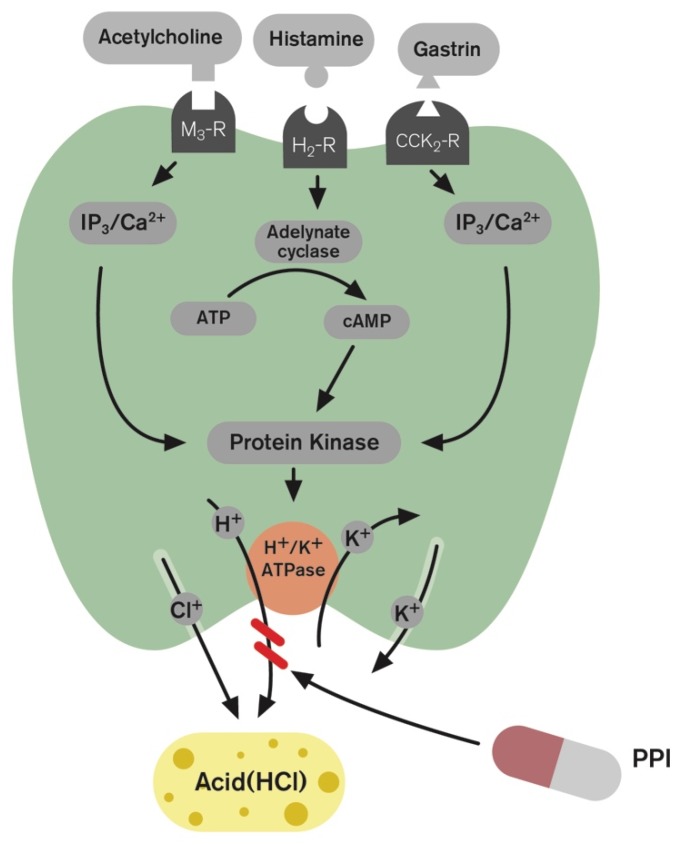
The parietal cells contain the H^+^/K^+^ ATPase or “proton pumps” located in the canaliculus of the parietal cell and responsible for the transport of acid (H^+^) into the stomach lumen. The main stimulants of acid secretion at the level of parietal cells are histamine, acetylcholine and to a lesser extent, gastrin. ATP, adenosine triphosphate; cAMP, cyclic adenosine monophosphate; CCK_2_-R, cholecystokinin type 2 receptor; H_2_-R, histamine type 2 receptor; IP_3_, inositol triphosphate; M_3_-R, muscarinic type 3 receptor; PPI, proton pump inhibitor.

**Figure 2 ijms-20-05469-f002:**
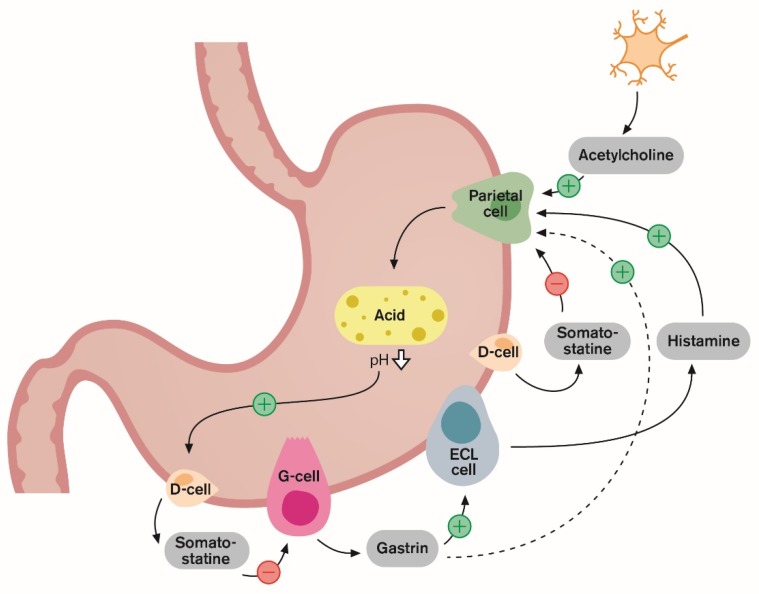
Protein in meals stimulate the G-cells to release gastrin into the blood. Gastrin stimulates the enterochromafin-like (ECL) cells to release histamine. The histamine then stimulates acid-producing parietal cells. This is the gastrin–ECL axis, the main stimulatory pathway of gastric acid secretion. The over-production of acid is prevented by negative feedback inhibition by intragastric acidity as low antral pH inhibits gastrin release via somatostatin from D-cells.

**Figure 3 ijms-20-05469-f003:**
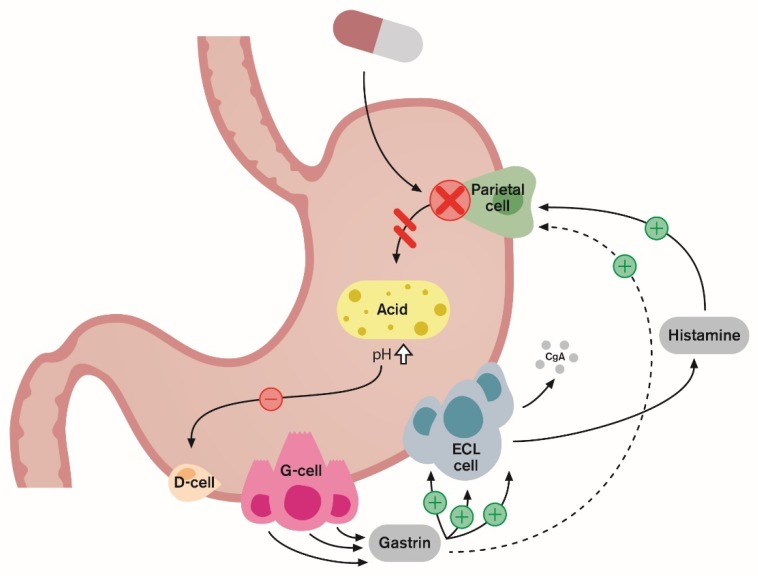
Protein pump inhibitors (PPIs) inhibit gastric acid secretion by binding covalently to active proton pumps on the parietal cells. This prevents acid secretion and leads to hypoacidity (higher pH level). Thereby somatostatin-mediated negative feedback of gastrin release on antral G-cells is inhibited, which leads to hypergastrinemia and gastrin exerts a trophic effect on the stomach’s mucosa, causing enterochromaffin-like (ECL) hyperplasia. Measurement of CgA levels in blood can be a useful tool for monitoring ECL cell hyperplasia secondary to treatment with PPIs.

**Figure 4 ijms-20-05469-f004:**
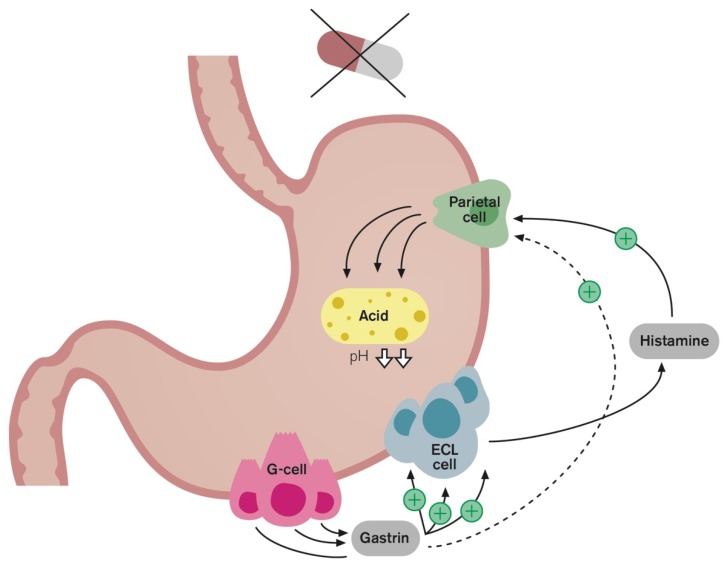
Following PPI discontinuation, the recovery of acid secretion can be exaggerated. Hypergastrinemia secondary to PPI therapy is associated with acid hypersecretion or the so-called rebound acid hypersecretion phenomenon. ECL cell, enterochromaffin-like cell.

**Figure 5 ijms-20-05469-f005:**
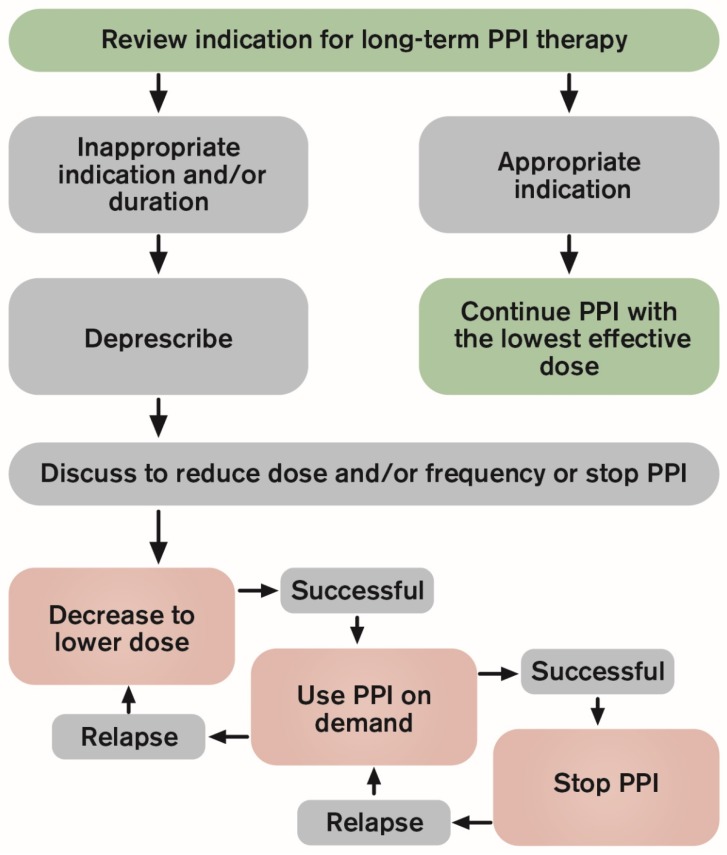
Different approaches in deprescribing PPIs.

**Table 1 ijms-20-05469-t001:** Steps in deprescribing proton pump inhibitors.

	The Steps of PPI Deprescribing
Step 1	Review indication and effectiveness
Step 2	Assess the balance of benefits and harms
Step 3	Assess patients values and preferences
Step 4	Decide wether to continue, reduce dose or discontinue PPI therapy
Step 5	Deprescribe and monitor

**Table 2 ijms-20-05469-t002:** Deprescribing studies that have elucidated success rate of different deprescribing methods and important factors associated with successful deprescribing or PPI requirement.

**Step-Down Studies**			
Authors: Helgadottir et al. (2017) [69]	Participants: GERD patients with EE (*n* = 50)	Deprescribing method: Step-down dose by half vs. continuous same-dose treatment	Outcome: Step-down was successful in 76%
Methods: A double-blind randomized trial Country: Iceland	PPI duration: > 2 years PPI dose: 40 or 20 mg Age: median 59 years Gender (F/M): 25/25	Setting: Hospital Follow-up time: 8 weeks	Comment: Female gender was an independent predictor for a successful step-down
Authors: Inadomi et al. (2003) [57]	Participants: Heartburn patients (*n* = 117)	Deprescribing method: Step-down from multiple- to single-dose	Outcome: Step-down was successful in 79.5%
Methods: A non-controlled prospective study Country: USA	PPI duration: > 8 weeks PPI dose: > 20 mg Age: median 66 years Gender (F/M): 5/112	Setting: VA hospital and outpatient clinic Follow-up time: 6 months	Comment: Longer PPI duration before step-down was an independent predictor of PPI requirement
**Discontinuing Studies**			
Authors: van der Velden et al. (2010) [47]	Participants: GERD patients on long-term PPI therapy (*n* = 141)	Deprescribing method: Abrupt discontinuation with 20 mg PPI as escape medication vs. daily 20 mg PPI with placebo escape medication	Outcome: 32% persisted daily PPI dosage, 43% reduced their dosage, 25% used less than 2 tablets/week.
Methods: A double-blind, parallel-group trial Country: The Netherlands	PPI duration: > 6 months PPI dose: 20 mg Age: mean 57 years Gender ratio: 56%	Setting: Primary care Follow-up time: 13 weeks	Comment: About 20% of long-term PPI users became satisfied on placebo with hardly any PPIs (0.7 tab-let/week)
Authors: Zwisler et al. (2015) [73]	Participants: Long-term PPI users without history of esophagitis, ulceration or current NSAIDs use. (*n* = 85)	Deprescribing method: Abrupt discontinuation vs. continuous treatment	Outcome: Discontinuation was successful in 27% of patients
Methods: A double-blinded randomised placebo-controlled trial Country: Denmark	PPI duration: > 8 weeks PPI dose: 40 mg Age: median 59 years Gender (F/M): 48/37	Setting: Primary care Follow-up time: 1 year	Comment: Significantly more men had an unsuccessful discontinuation
Authors: Björnsson et al. (2006) [11]	Participants: Long-term PPI users without PUD or EE (*n* = 96)	Deprescribing method: Discontinuation: abrupt vs. 3 weeks tapering	Outcome: Discontinuation was successful in 27% (31% of tapering and 22% of abrupt discontinuation, NS)
Methods: A double-blind, placebo-controlled trial Country: Sweden	PPI duration: > 8 weeks PPI dose: 20 mg Age: median 63 years Gender (F/M): 52/44	Setting: Hospital Follow-up time: 1 year	Comment: GERD and serum gastrin were independent predictors of PPI requirement
Authors: Pilotto et al. (2003) [46]	Participants: Erosive esophagitis patients (*n* = 56)	Deprescribing method: Abrupt discontinuation vs. continuous treatment	Outcome: 62.5% had a relapse of erosive esophagitis 6-months after discontinuation
Methods: A prospective, randomized, double-blind study Country: Italy	PPI duration: 6 months PPI dose: 20 mg Age: > 65 years of ageGender (F/M): Not given for the double-blind phase	Setting: Hospital Follow-up time: 1 year	Comment: 81% healing rate was in the maintenance phase after 4-months of 20 mg following a step-down from 40 mg for 8 weeks.
**On-Demand Studies**			
Authors: Bayerdorffer et al. (2016) [66]	Participants: Symptomatic NERD patients (*n* = 301)	Deprescribing method: On-demand vs. continuous treatment	Outcome: On-demand was successful for 92%
Methods: A multicenter, open-label, randomized, parallel-group study Country: Austria, France, Germany, South Africa and Spain	PPI duration: 4 weeks PPI dose: 20 mg Age: mean 48 years Gender (F/M): 179/122	Setting: Hospital Follow-up time: 6 months	Comment: On-demand treatment was non-inferior to continuous treatment
Authors: Bour et al. (2005) [67]	Participants: Non-severe GERD patients with frequent symptom relapses (*n* = 71)	Deprescribing method: On-demand vs. continuous treatment	Outcome: On-demand was successful, with a high symptom relief in 74.6%
Methods: A randomized, open-label study Country: France	PPI duration: > 1 year PPI dose: 10 mg Age: average 50 years Gender ratio: 58% men	Setting: Hospital Follow-up time: 6 months	Comment: There was a significant decrease in medication consumption in the on-demand group
Authors: Janssen et al. (2005) [68]	Participants: GERD patients (*n* = 215)	Deprescribing method: On-demand vs. continuous treatment	Outcome: On-demand was successful in 69.3%
Methods: A multicentre, open-label Country: Germany, France, Switzerland and Hungary.	PPI duration: 4 weeks PPI dose: 20 mg Age: mean 50 years Gender (F/M): 115/100	Setting: Did not describe the clinical settings or type of centers Follow-up time: 6 months	Comment: Patients were satisfied with the on-demand therapy which was non-inferior to continuous therapy with regard to symptom control

Note: The number of study subjects are given for the size of the deprescribing group of the studies. Abbreviations: EE, erosive esophagitis; F, female; M, male; NERD, non-erosive reflux disease; NS, non-significant; NSAIDs, nonsteroidal anti-inflammatory drugs; PPI, proton pump inhibitors; PUD, peptic ulcer disease; USA, United States of America; VA, Veterans Affairs.

**Table 3 ijms-20-05469-t003:** List of indications for long-term proton pump inhibitor therapy.

Indications for Continuous PPI Therapy
Severe esophagitis (LA grade C or D) Barrets esophagus
Documented history of bleeding GI ulcer
Chronic NSAIDs use with bleeding risk factors
Zollinger-Ellison syndrome

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
