# Peer review of "Problems Associated with Deprescribing of Proton Pump Inhibitors"

_ijms, 2019, doi:10.3390/ijms20215469_

Round 1

Reviewer 1 Report

The review nicely summarizes the current understanding of PPI treatment and associated long term safety profile. The important topic of PPI deprescribing is very relevant in everyday clinical practice.

The following issues should be addressed:

Figure 1 does not contain much information and could be improved by a more detailed depiction of the mechanism of acid secretion.

Chapter 2 including the four categories of PPI side effects could be structured in a clearer fashion with subchapters.

Is there evidence on the clinical effectiveness of tapering PPI therapy in reducing rebound symptoms?

Table 2 needs reformatting to clearer show the connection between the reference and related information.

Chapter six should focus on the current recommendations on PPI deprescribing and not on the limitations of the available evidence which is addressed in the next chapter. An accessible summary of recommendations including the authors expert opinion in the best practice of PPI deprescribing should be included.

Author Response

I am very thankful for the positive words that the reviewer has to say about our review.

Figure 1 does not contain much information and could be improved by a more detailed depiction of the mechanism of acid secretion.

Response: More details have been added to Figure 1, the intracellular second messenger pathways of acid secretion as wells as the transport of ions into the lumen of the stomach. The events leading to acid secretion by the parietal cell are also described in more detail in chapter 1: “[…]by binding to M3 receptors. It is now recognised that the stimulation of acid secretion by gastrin mainly occurs via histamine release from the ECL cells, the gastrin–ECL axis. Histamine from the ECL cells is considered the limiting step in stimulating maximal gastric acid secretion [5]. Gastrin/CCK2 receptors have been well documented on ECL cells but not on parietal cells [6]. However, gastrin is believed to stimulate parietal cells directly, to some extent, but this is considered to be a less extensive mechanism [1]. The stimulatory effect of acetylcholine and gastrin is mediated by an increase in cytosolic calcium (Ca2+), whereas that of histamine is mediated by the activation of adenylate cyclase and generation of cyclic adenosine monophosphate (cAMP) [2]. The second messengers (cAMP and Ca2+) activate protein kinases and phosphorylation cascades that ultimately activate transport of the proton pump (H+/K+ ATPase) from cytoplasm to the canaliculus where the pump can exchange intracellular H+ with the extracellular K+, which represents the gastric  acid secretion [2] (Figure 1). Therefore, a combination of the effects of these stimulants can augment the gastric acid secretion.”

Chapter 2 including the four categories of PPI side effects could be structured in a clearer fashion with subchapters.

Response: This issue has been addressed by subchapter headlines for each category (2.1 – 2.4): “The putative side effects of PPI therapy can be divided in to four categories based on the effects and metabolism of this drug class:

2.1) Hypochlorhydria: Side effects related to the direct effect of acid suppression or PPI-induced hypochlorhydria has been associated with an increased risk of infections [23-28], malabsorption of nutrients, minerals and vitamins and increased risk of osteoporotic fractures [4,25].

2.2) Idiosyncratic: Idiosyncratic side effects, are rare and as the term implies are unpredictable among PPI users. These include, renal failure (i.e., acute interstitial nephritis) [29] and the association with cardiovascular disease [30] and dementia [31].

2.3) Pharmacokinetic (PK) interaction: The PK interaction is related to the effects on the cytochrome P450 (CYP450) cytochromes and possible drug interactions when prescribed with other drugs. There is lack of convincing clinically important drug interactions with PPIs, the most studied combination is the use of clopidogrel with PPIs [32] which has been associated with higher adverse cardiovascular events, however results have been mixed and it is unclear if there is a true difference in clinical outcomes [33].

2.4) Hypergastrinemia. The clinical implications of PPI-induced hypergastrinemia are also of concern. These concerns are largely related to […]”

Is there evidence on the clinical effectiveness of tapering PPI therapy in reducing rebound symptoms?

Response: Currently there are no data to support the hypothesis that tapering reduces rebound symptoms. And this is mentioned in chapter five and again in chapter 7: “In one Swedish study, tapering was conducted over a period of three weeks before discontinuation and compared with abrupt discontinuation [11]. Tapering was not shown to be superior for the successful discontinuation of PPI therapy, 31% in the tapering group vs. 22% in the non-tapering group were off PPIs at 1-year follow-up [11].” & Although no evidence that PPI step-down before discontinuation is better than abrupt discontinuation [11], tapering is believed to be more effective [61] and this strategy for discontinuation is recommended in the Australian algorithm [44]. However, this recommendation is mostly based on physiological studies [52,53] since there is limited data on PPI withdrawal in patients and with conflicting results [57,56].”

Table 2 needs reformatting to clearer show the connection between the reference and related information.

Response: We believe Table 2 clearly shows the connection between the reference (author, year and reference number are given in the table) and the related information. We have changed the shading to separate each study. However we are pleased to change table 2 but we are not certain what the reviewer means.

Chapter six should focus on the current recommendations on PPI deprescribing and not on the limitations of the available evidence which is addressed in the next chapter. An accessible summary of recommendations including the authors expert opinion in the best practice of PPI deprescribing should be included.

Response: Chapter six has been changed according to the reviewer’s suggestions, and recommendations from the authors have been added: “The authors would like to suggest two important practice point when prescribing PPIs, the preventive approach and the detective approach. The preventive approach is when empiric PPI trial is used as a diagnostic indicator for GERD in patients with typical reflux symptoms in the absence of alarm features. The trial should be a short PPI course, we believe the 4 – 8 weeks treatment approach that is often prescribed is more than needed to assess treatment response. It is estimated that the daily dosing of PPIs reaches a steady state of inhibition after five days, and that state is the inhibition of about 66% of the maximal acid output [3]. Therefore, a short 1-week PPI course should be enough to assess response. There is need for randomized trials to support this but this approach is likely to avert overutilization and minimize possible withdrawal symptoms in patient who do not benefit from PPI therapy. The detective approach is when renewing PPI prescriptions for patients already on PPI therapy. Then the steps listed in Table 1 should be used to assess the patients need for continued PPI therapy and candidates for deprescribing should subsequently attempt to reduce dose and/or frequency of PPIs or stop them. The success and relapse of the deprescribing approaches listed in Figure 5 should be periodically reevaluated so that the lowest effective PPI dose is used. This approach is likely to lower medication costs and lower the risks of long-term side effects among GERD patients prescribed maintenance PPI therapy for symptom control.” And this part was moved to chapter 7: Due to the paucity of publised deprescribing trials, the current evidence is generally of low methodological quality and provides relatively low-certainty evidence. Randomized controlled trials (RCT) are considered to be the gold standard when it comes to high level of evidence. However, RCTs are not always applicable when estimating the safety and effectiveness of drug deprescribing, as in the case of PPIs, when the adverse effects are rare or take a long time to develop. Furthermore, the small sample size often seen in the RCTs tend to be relatively homogeneous.The guidelines published are mainly based on studies on patients with moderate GERD or mild esophagitis. Thus, they are not always adequately representative of the real-world population of PPI users.”

Reviewer 2 Report

The review of Helgadottir and Björnsson on problems associated with deprescribing of PPIs is written meticulously. It summarize current state of knowledge on the reviewed topic. However, what I am missing in the manuscript is a brief description of molecular biology and physiology of the proton pump and a bit more details about interactions between PP and PPIs at the molecular level.

Overall the manuscript should be accepted for the publication in MDPI after minor corrections.

Author Response

I am very thankful for the positive words that the reviewer has to say about our review.

Response: The interaction between PP and PPI have been added to chapter 1: “The H+/K+ ATPase has been shown to be an α, β-heterodimeric enzyme which catalyses a one-to-one exchange of hydrogen (H+) and potassium (K+) ions [2,3].” & “PPIs are weak bases and therefore accumulate in the acidic space, the secretory canaliculus, of parietal cells [3]. After acid-induced activation PPI covalently bind to the active proton pump (H+,K+-ATPase), the binding is achieved through the disulphide bond between the activated PPI and cysteines of the pump enzyme [3]. This covalent binding enables the prolonged inhibition of acid secretion, even after the drug concentration in blood has waned. The duration of the inhibitory activity is variable and affected by pump turnover and the loss of covalently bound PPIs [8]. PPIs are most effective when the parietal cells are stimulated to secrete acid, as they are after a meal; as such, PPIs should be administered before a meal [9].”